# A 479-Year Early Summer Temperature Reconstruction Based on Tree-Ring in the Southeastern Tibetan Plateau, China

**Yu Zhang** [1] , **Jinjian Li** [1,*] , **Zeyu Zheng** [2] **and Shenglan Zeng** [1]

1   Plateau Atmosphere and Environment Key Laboratory of Sichuan Province, School of Atmospheric Sciences, Chengdu University of Information Technology, Chengdu 610225, China; yu.z2020@foxmail.com (Y.Z.); zengsl@cuit.edu.cn (S.Z.)

2   MOE Key Laboratory of Western China's Environmental System, College of Earth and Environmental Sciences, Lanzhou University, Lanzhou 730000, China; zhengzy17@lzu.edu.cn

\*   Correspondence: ljj@cuit.edu.cn

**Abstract:** Due to the lack of long-term climate records, our understanding of paleoclimatic variability in the Tibetan Plateau (TP) is still limited. In this study, we developed a tree-ring width (TRW) chronology based on tree-ring cores collected from our study site, southeastern TP. This chronology responded well to the mean maximum temperatures of May–June and was thus used to reconstruct early summer (May–June) maximum temperature during the period 1541–2019. The reconstruction explained 33.6% of the climatic variance during the calibration period 1962–2019. There were 34 extremely warm years (7.2% of total years) and 36 extremely cold years (7.5% of total years) during the reconstruction period. The spatial correlation analysis and the comparison with other local temperature reconstructions confirmed the reliability and representativeness of our reconstruction. The results of the ensemble empirical mode decomposition (EEMD) analysis indicated quasi-oscillations of 2.9–4.2 years, 4.5–8.3 years, 11.1–15.4 years, 20–33.3 years, 50.4 years, 159.7 years, and 250 years in this temperature reconstruction which may be associated with ENSO cycles, solar activity, and PDO.

**Keywords:** tree-rings; climate variability; southeastern Tibetan Plateau; global sea surface temperature

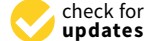



## 1. Introduction

Global warming has significant impacts on the forest ecosystem, natural resources, and social economy, such as through land degradation, landslides and the retreat of glaciers [1]. The Tibetan Plateau (TP), one of the most sensitive regions to climate change, plays a crucial role in driving large-scale atmospheric circulations in the Northern Hemisphere by heating the troposphere [2–4]. Thus, exploring climate variations in this region over the past century and its evolution under the background of global warming is of vital importance to predict future extreme events and make relevant strategies in its downstream areas. However, meteorological stations over the TP are sparsely distributed and existing instrumental climatic records are relatively short, which restricts our ability to evaluate low-frequency climate variations. To improve our understanding of regional climate change from a long-term perspective, it is necessary to develop another high-resolution paleoclimatic reconstruction. Tree ring, with the characteristics of wide distribution, high resolution and high sensitivity to climate, is an important information carrier of climate change and has a unique position in climate research [5–7]. In recent decades, climate reconstructions based on tree-ring data originating from the TP have been widely developed, including temperature [8–13], precipitation/drought [14–17] and river runoff [18–20]. These climate reconstructions inferred from tree-ring show us regional information of climate variability on the TP. The southeastern TP, which belongs to the so-called subtropical mountain canyon area, has most of its areas controlled by monsoon circulation [21]. Therefore, tree-ring studies conducted in the southeastern TP may provide essential information about paleoclimatic variability in this region and may further improve our understanding of large-

scale atmospheric circulation systems which can affect regional and even global climate variations. For example, Keyimu et al. [21] reconstructed a 413-year-long annual minimum temperature series on the southeastern TP. Yu et al. [22] developed a summer temperature variability reconstruction for the central Hengduan mountains, southeastern TP, during the period 1795–2008. Duan and Zhang [23] reported the reconstruction of the April–September mean temperature over the past 449 years on the southeastern TP and explored its relation to solar activity. All of these studies found the possible links between the tree-ring growth and large-scale atmospheric circulation systems. However, a few of the tree-ring studies are situated in the northern part of the Shaluli Mountains, southeastern TP. The temperature series reconstructed in this study area can provide important updates and increased data point density to the tree ring proxy network of the northern Hemisphere, and further our understanding of temperature change. Here, we presented a chronology, originating from Xinlong county, on the southeastern TP. As shown below, the high sensitivity of the tree-ring width (TRW) chronology to early summer maximum temperature allows us to develop a reliable reconstruction to perceive regional early summer maximum temperature variations during the past five centuries and to investigate the possible driving factors that influence the temperature variability on the southeastern TP.

## 2. Materials and Methods

### 2.1. Study Area

Our study area is located in Xinlong (30°81′ N, 100°35′ E, 3880 m a.s.l.) on the southeastern TP (Figure 1). The monsoon systems of the Indian and Pacific Oceans dominate this area in the summer months, which transport warm, moist air from the Bay of Bengal to the southeastern TP; while during winters, continental air masses (Asian winter monsoon) dominate and lead to dry and cold conditions in this area [8,24,25]. The climate data from 3 meteorological stations close to our tree-ring sampling sites during the period 1962–2019 show that the annual mean temperature was 6.7 °C and the multi-year mean of annual total precipitation was 601.1 mm, of which more than 80% of the precipitation occurs between May and September. The monthly average maximum and minimum temperatures were 16.0 °C and 0.1 °C, respectively (Figure 2).

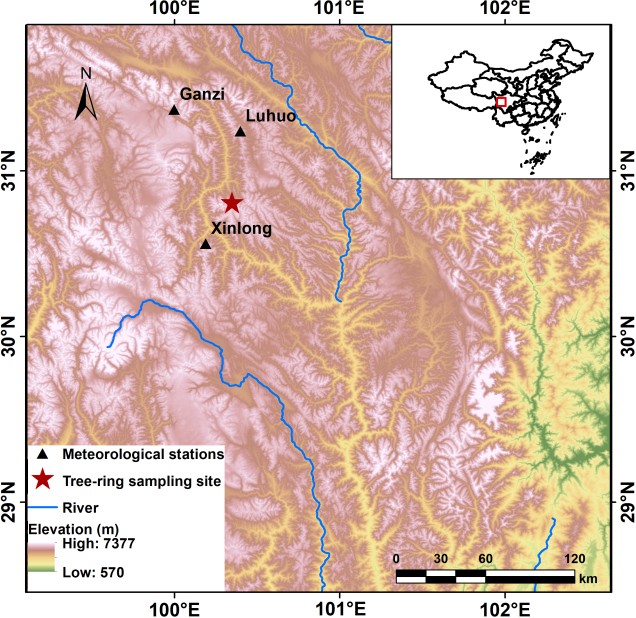

**Figure 1.** Tree-ring sampling site (red pentagram) and three meteorological stations (black triangle) in the southeastern Tibetan Plateau.

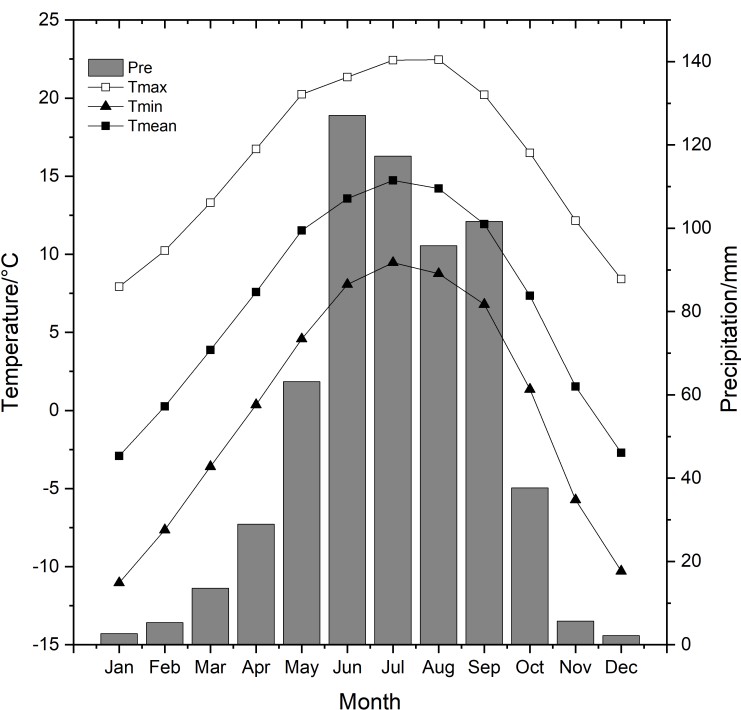

**Figure 2.** Monthly mean (Tmean), maximum (Tmax), minimum (Tmin) temperatures, and monthly total precipitation (Pre) calculated using climate data from 3 meteorological stations (Ganzi, Luhuo, and Xinlong) around our sampling site during the period 1962–2019.

### 2.2. Climate Data

Monthly maximum temperature (Tmax), mean temperature (Tmean), minimum temperature (Tmin), and precipitation (Pre) data were obtained from 3 meteorological stations (Ganzi [100° E, 31°62′ N, 3394 m a.s.l.], Xinlong [100°32′ E, 30°93′ N, 3000 m a.s.l.], and Luhuo [100°67′ E, 31°40′ N, 3250 m a.s.l.]) near to our tree-ring sampling site (Figure 1). To investigate the climate–growth relationship of the tree-ring samples with climate variables, the averaged data from the 3 meteorological stations during the period 1962–2019 were selected to perform climate–tree growth relationship analyses. The monthly 0.5° × 0.5° gridded temperature dataset from CRU TS 4.04 [26] was used to assess the spatial representation of the reconstruction. The 2° × 2° gridded NOAA Extended Reconstructed Sea Surface Temperature (SST) V5 data also was used to explore the relationship between the study site and global SST [27].

### 2.3. Tree-Ring Data and Chronology Development

Tree-ring samples were collected at 3880 m a.s.l. in August 2019 from *Spruce-Fir* mixed stands in Xinlong (Figure 1). Only *Picea balfouriana* was sampled. The sampled trees were growing on 30–40% slopes and were far from areas of anthropogenic activity and disturbance. Increment cores were obtained from each selected healthy tree by an increment borer at breast height (1.3 m).

In the laboratory, after air drying, all selected increment cores were carefully glued to wooden mounts, then sanded with progressively finer sandpaper. After that, each core was cross-dated to specify the exact calendar year for each tree ring and then all tree-ring widths were measured with 0.01 mm precision using the software Measure J2X. Finally, to verify the accuracy of crossdating and measurements, the program COFECHA was used [28]. Sample cores with much abnormal ring features, such as missing rings, which are difficult to cross-date, were excluded. Eventually, 46 cores from 31 trees were used to develop a ring-width chronology with a missing tree-ring rate of 0.068% (7 samples with missing rings).

The growth of trees is not only is affected by environmental factors but is also affected by non-climatic disturbances and age-related factors. These effective factors needed to be removed for dendroclimatic study. This procedure is referred to as "tree-ring standardization" [29]. The ARSTAN program was used to perform "tree-ring standardization" from the raw data [30]. To find the best detrend method for this study, all series were detrended by several methods—including smoothing spline, linear regression, and negative exponential curve. After comparing all the detrend methods, the 50 years of series length smoothing spline was selected. Finally, the detrended index series were combined into a single TRW chronology by calculating a biweight robust means chronology [31] over a 775-year period spanning AD 1245 to 2019.

The signal strength of the chronology was evaluated by employing the expressed population signal (EPS) and running average correlation between series (RBAR) [29]. The accepted threshold was EPS > 0.85 [32]; thus, we used the EPS with a threshold value of 0.85 to assess the most reliable time span of this chronology. The threshold of 0.85 was reached at a sample depth in 9 cores in our study (Figure 3) and we thus regard the most reliable period to be 1541–2019. The following analyses were based on the reliable period for 1541–2019.

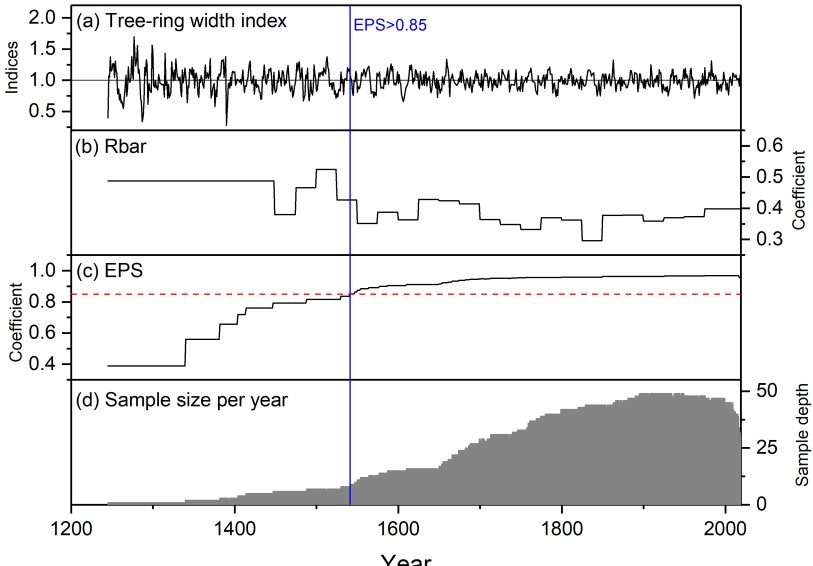

**Figure 3.** (**a**) Tree-ring width index; (**b**) inter-series correlation (Rbar) for the standard chronology; (**c**) expressed population signal (EPS) (dashed red line indicates 0.85 threshold); and (**d**) the sample size (gray shadow) changes over time. Critical EPS level of 0.85 is highlighted with vertical blue line.

*2.4. Climate Reconstruction and Statistical Methods*

To identify the key climatic factors for tree growth in this study area, Pearson's correlation analysis was conducted between TRW chronology and four climate variables (mean temperature, maximum temperature, minimum temperature, and precipitation). Then, a transfer function was established using the linear regression method with the selected climate variable and TRW chronology. To test the stability and quality of the regression model, split-period calibration/verification analysis [29,31] was employed. Split-period calibration/verification analysis provides statistics including Pearson's correlation coefficient (R), explained variance ($R^2$), adjusted explained variance ($R^2_{adj}$), the sign tests of both the original data (ST) and first-differenced data (ST1), the coefficient of efficiency (CE), and reduction of error (RE) [29,31]. In addition, spatial field correlation analysis between the reconstructed climate series and the 0.5° × 0.5° gridded Climatic Research Unit (CRU) TS 4.04 dataset [26] was employed to investigate the spatial consistency of the reconstructed series during the period 1962–2019. Then, our reconstructed series was compared with other nearby reconstructions and glacier fluctuations to verify the reliability

of our reconstruction. Ensemble empirical mode decomposition (EEMD) [33,34] was used to provide multi-scale decomposition of the reconstruction. This adaptive time–frequency method of data analysis can decompose any complex signals into a sequence of isolated intrinsic mode functions (IMFs) with different time scales.

## 3. Results

### 3.1. Climate–Tree Growth Response Analysis

The results of correlation analyses between the TRW chronology and four monthly climate variables during the period 1962–2019 were demonstrated in Figure 4. According to the results, the correlations between TRW chronology and maximum/mean/minimum temperature showed a significant positive correlation with current June ($p < 0.01$). In addition, the chronology also showed significant correlations with the maximum/mean temperature of current May and July ($p < 0.05$). For precipitation, the significant correlation ($p < 0.05$) only occurred in July of the current year. These results suggested that the temperature is the main controlling factor for the tree-ring change in our study area. Since the seasonally averaged climate factors are often more representative of long-term climatic conditions than individual months, the best seasonal factor for climate reconstruction was further evaluated. As a result, the strongest correlation was found between the TRW chronology and mean maximum temperatures of May–June (r = 0.579, $p < 0.01$). Therefore, we selected the mean maximum temperatures of May–June (early summer) as the reconstructed factor in this study.

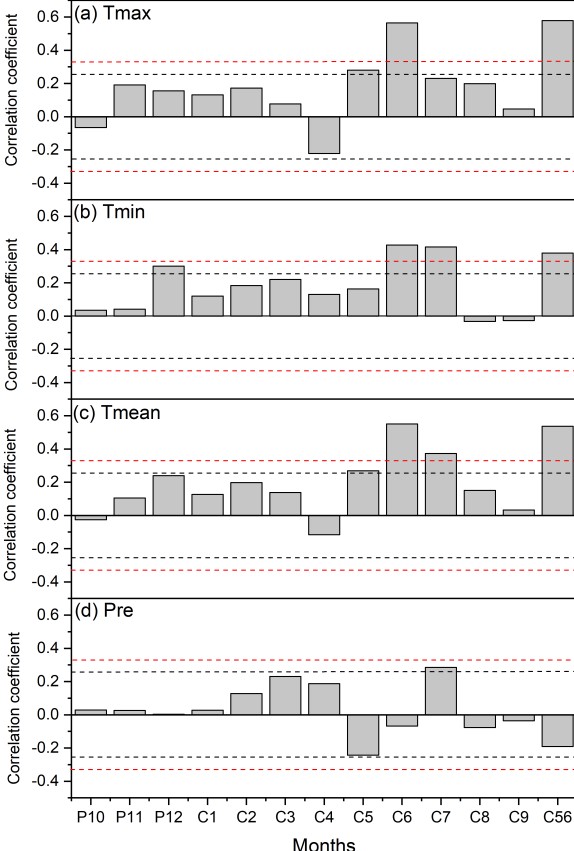

**Figure 4.** Correlation coefficients between the TRW chronology and the monthly maximum (**a**), minimum (**b**), mean temperature (**c**) and monthly total precipitation (**d**) during the period 1962–2019. P marks the previous year and C marks the current year. The label C56 denotes the average during the period May–July. The horizontal red/black dotted lines indicate a statistical significance level of 0.01 and 0.05, respectively.

### 3.2. Calibration and Verification of the Reconstruction Model

Based on the linear regression method, we reconstructed the mean maximum temperatures of May–June during the period 1541–2019. The reconstruction model accounted for 33.6% ($R^2_{adj}$ = 32.4%, F = 28.30) of the temperature variance during the calibration period 1962–2019. The split-period calibration–verification result is shown in Table 1. During the verification period, the ST passed the significance test at 0.05 level and RE was 0.29. These results indicated the good performance of the regression model in reconstructing the mean maximum temperatures of May–June. The ST1 and CE did not pass the significance test. Similar results have also appeared in previous studies [35,36], which may be attributed to the great temperature difference between the calibration period and the verification period. However, from the test results of the whole reconstruction period, the equation was relatively stable and reliable. Consequently, the equation (Y = 5.316 × TRWC + 15.582; where Y is the mean maximum temperatures of May–June) during the period from 1962 to 2019 was considered reliable and was selected to reconstruct the mean maximum temperatures of May–June variations back to 1541.

**Table 1.** Statistics of the split calibration–verification of tree-ring reconstruction of mean maximum May–June temperature.

| Calibration | | | | | Verification | | | | | |
|---|---|---|---|---|---|---|---|---|---|---|
| Period | r | $R^2$ | $R^2_{adj}$ | F | Period | r | ST | ST1 | RE | CE |
| 1962–1990 | 0.574 | 0.330 | 0.305 | 13.30 | 1991–2019 | 0.699 | 22+/7− | 17+/11− | 0.267 | −0.03 |
| 1991–2019 | 0.699 | 0.489 | 0.470 | 25.79 | 1962–1990 | 0.574 | 20+/9− | 17+/11− | 0.103 | −0.20 |
| 1962–2019 | 0.579 | 0.336 | 0.324 | 28.30 | | | | | | |

### 3.3. Variations of Mean Maximum May–June Temperature

Using the regression model above, we reconstructed the maximum May–June temperature for the southeastern TP during the period 1541–2019 (Figure 5b). We also compared the reconstructed early summer temperature series with the actual records during the instrumental period, finding them to be consistent at both higher and lower temperatures (Figure 5a). The mean of the reconstructed temperature was 20.85 °C and the standard deviation (SD) of the reconstructed series was 0.62 °C. An extremely warm summer was identified as at least 1.5 SD above the mean, while an extremely cold summer was defined as at least 1.5 SD below the mean. According to our reconstruction, there were 34 extremely warm summers (7.2% of total summers) and 36 extremely cold summers (7.5% of total summers) during the reconstruction period. The warmest summer (1659) was 22.67 °C and the coldest summer (1606) was 19.09 °C. Extremely warm periods (continuously above mean + 1.5 SD for more than 2 years) were 1562–1564, 1614–1616, 1659–1661, and 1783–1785, while extremely cold periods were 1567–1573, and 1604–1608. No extremely warm or cold periods were found during the most recent 200 years. Based on the 11-year low-pass filter, there were 13 warm periods (defined as a period with continuously above-average temperature for 11 years) and 14 cold periods (defined as a period with continuously below-average temperature for 11 years). Furthermore, there was no apparent ascending tendency displayed in the maximum May–June temperature for the past 50 years compared to the values for the preceding periods.

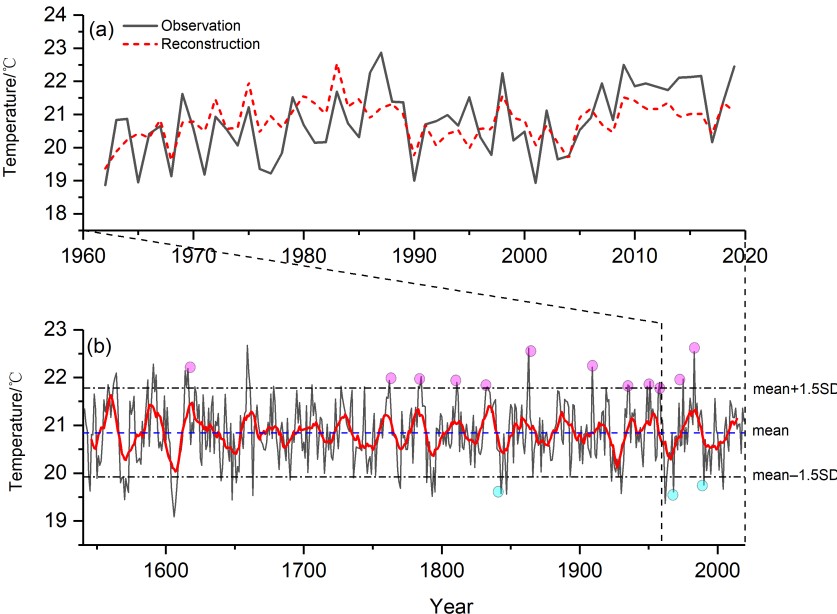

**Figure 5.** Comparison between the reconstructed and observed temperature during the period 1962–2019 (**a**); the reconstructed mean maximum temperatures in May–June using the reliable period of the TRW chronology from 1541 to 2019 with historical extreme cold (blue circle) and the extreme dry (red circle) events corresponding to it (**b**). The red line in (**b**) indicates the 11-year low-pass filtered time series, and the blue dashed horizontal line in (**b**) denotes the mean value and the black horizontal dashed lines represent the standard deviation value.

## 4. Discussion

### 4.1. Climate–Radial Tree Growth Relationships

Previous studies showed that temperature is a critical factor limiting tree growth on the TP. For example, Yin et al. [12] found the tree-ring growth is influenced by the warm-season (April–September) temperature on the eastern TP. Chen et al. [37] found that the tree-ring chronology was significantly negatively correlated with April–June maximum temperature, indicating that maximum temperature is the main factor that limits tree growth in Animaqin Mountains of the TP. Li et al. [38] found late summer (August–September) temperature exerted considerable influence on the tree growth in the Gaoligong Mountains, southeastern TP. In this study, the mean maximum temperature of early summer is the key climatic factor influencing radial tree growth at higher elevations on the southeastern TP. Located in a semi-humid area of TP, the precipitation total in May and June was 63.2 mm and 127.0 mm in our study area, respectively. Early summer is the vigorous period of tree growth in the southeastern TP, with abundant rainfall—where a higher temperature means more conducive photosynthesis and cambium cell division as well as the formation of wider rings; conversely, if the temperature is lower, it will reduce the photosynthetic efficiency and form narrow rings. Therefore, the radial growth of *Picea balfouriana* in southeastern TP is significantly positively correlated with the maximum May–June temperature, which has a clear tree physiological significance.

### 4.2. Validation for Our Summer Temperature Reconstruction

#### 4.2.1. Spatial Correlation Analysis

Spatial correlations between the reconstructed mean maximum temperature of May-June and CRU gridded data (TS 4.04) during the period 1962–2019 was displayed in Figure 6a, which shows a similar pattern with the result of actual concurrent gridded data (Figure 6b). To some extent, these results revealed that the temperature reconstruction could represent the early summer maximum temperature variability of the study area

as well as the entire TP. The correlation of reconstructed records was not as strong as that of instrumental records. This difference may be due to the loss of variance in the reconstruction model.

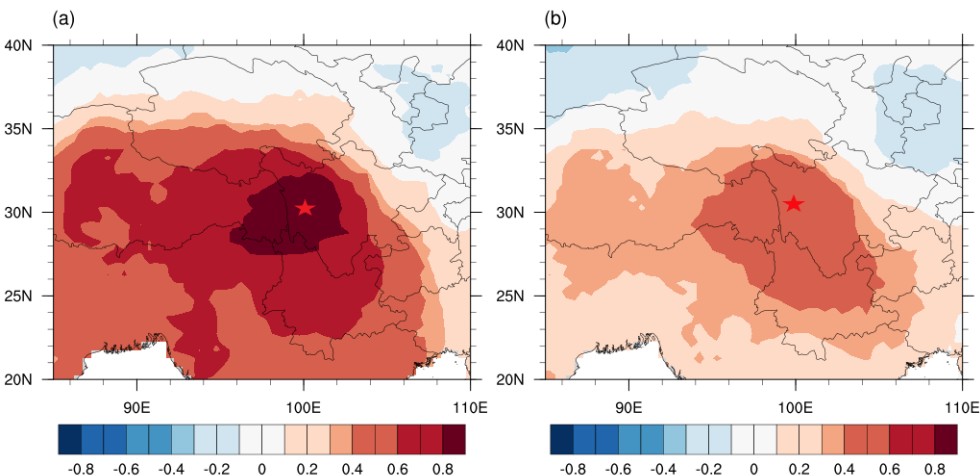

**Figure 6.** Spatial correlations of (**a**) observed and (**b**) reconstructed mean maximum May–June temperatures for the southeastern Tibetan Plateau with gridded June–July temperature from Climatic Research Unit (CRU) TS 4.04 during the period 1962–2019. The pentagram indicates the sampling site.

### 4.2.2. Comparison with Nearby Temperature Reconstructions

To assess whether the temperature reconstruction in this study represents similar features that were coherent over a large spatial scale on the southeastern TP, it was compared with other temperature reconstructions from surrounding temperature sensitive sites, including the annual mean temperature reconstructed in the southeastern TP by Keyimu et al. [21], mean summer (June–August) minimum temperature reconstructed in the source region of the Yangtze River by Liang et al. [25], minimum winter (November–February) temperature reconstructed on the southeastern TP by Huang et al. [24] and April–September temperature reconstruction for the central Hengduan Mountains by Fan et al. [10]. To compare the low-frequency signals in these temperature reconstructions, an 11-year low-pass filter was applied on each of the reconstruction series. It showed that all reconstructions displayed the similarity of warm/cold periods between these different reconstructions (Figure 7). As shown in Figure 7, the temperature reconstruction in this study and reconstruction by Huang et al. [24] exhibited the same warm periods in 1550–1560 and 1581–1595; while the same cold periods were experienced in 1566–1580. In addition, 1640–1650 (except in Huang et al. [24] and Fan et al. [10]), 1760–1775, 1820–1830 (except in Fan et al. [10]), 1860–1880, and 1900–1915 intervals experienced notable cold periods; while the 1780–1790, 1831–1845, 1881–1899, 1935–1955, and 2000–2014 intervals experienced notable warm periods. There was an upward warming trend in our reconstruction after the 20th century, which is less significant than other reconstructions. We further explored the reason why this temperature reconstruction did not show a significant warming as other reconstructions (Figure 7b,d) did. Firstly, the discrete warming trend in the May–June maximum temperature over the past 479 years in our study might be related to the topography and vegetation cover of the region under study. Xinlong is located in an area near Yalong River, and the elevations of our sampling sites are high and close to treelines. It has been proven that treelines on the southeastern TP hold much moisture in the rainy season [39]. This abundant moisture is favorable to mitigate the potential warming by evapotranspiration [40]. Secondly, the insignificant warming trend could be attributed to different targets of reconstruction (month/seasons), the use of different types of tree-ring proxies (width/density), and differences among seasons and species [21]. This insignificant warming trend in summer was also reported in other previous studies from the southeastern TP [41,42].

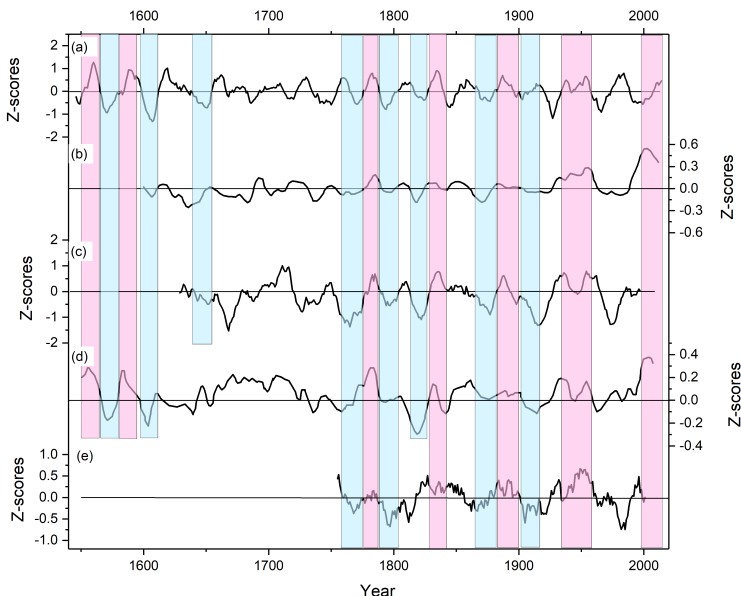

**Figure 7.** Comparison of the reconstructed mean maximum May–June temperature with four tree-ring-based temperature reconstructions near our study area. (**a**) The reconstructed mean maximum May–June temperature from this study; (**b**) the annual mean temperature reconstructed in the southeastern TP [21]; (**c**) the mean summer (June–August) minimum temperature reconstructed in the source region of the Yangtze River [25]; (**d**) the reconstructed minimum winter (November–February) temperature on the southeastern TP [24]; (**e**) the April–September temperature reconstruction for the central Hengduan Mountains [10]. All series were standardized and the black bold lines denote the 11-year FFT smoothing. Red and blue vertical bars represent the warm and cold periods, respectively.

### 4.2.3. Glacier Fluctuations

As previous studies have shown, advance or retreat phases of monsoonal–temperate glaciers are largely determined by the changes in air temperature, especially during summer temperature [25,43]. Therefore, we compared the historical glacier fluctuations in the surrounding regions with our reconstruction to further verify the accuracy and stability of the reconstruction sequence. The cold period 1955–1970 in this reconstruction corresponded to the glacier advance from 1957 to 1982 in the Baishui number 1; while the warm period 1940–1955 corresponded to the glacier retreat from 1932 to 1959 in the Melang glacier [44]. The Hailuogou glacier in the Gongga Mountain of the eastern TP was static or advanced during the early twentieth century (1900–1930), and was in a retreat state during the periods 1930–1966 and 1981–1989 [45]. These coincide with the cold period and warm period in this reconstruction, respectively. In addition, a conspicuous period with high temperatures during the 1770s in this new reconstruction was approximately concurrent with the beginning of a retreat for the Midui glacier in the southeastern TP [38,46]. Therefore, the reconstructed mean maximum temperature of May–June variability in the southeastern TP may be used as an indicator of past glacier fluctuations in nearby areas.

### 4.2.4. Historical Events

Furthermore, we compared the reconstructed temperature series with the extreme disaster events recorded in a calamity memorandum in Sichuan [47], China, finding that there were corresponding historical documents near the sampling site (Figure 5b). The documental records displayed an extreme cold May in 1843 and 1990 and persistent low temperatures from March to July in 1968, which caused damage to a large number of crops and led to production reduction. These records were consistent with the corresponding extremely cold summers in our reconstruction. Compared with the detailed records of extreme low temperature events, there were almost no direct records of high temperature events. In some previous studies, the drought events around the sampling points have

been used to verify the high temperature events [48]. There were many drought events recorded in the calamity memorandum in Sichuan before 1900. Among them, drought events were recorded in 1621, 1761, 1783, 1785, 1811, 1833, and 1863 near the sampling site. Correspondingly, these years exhibited an extremely warm summer in the temperature reconstruction. In addition, there were abnormally severe droughts during 1919, 1935, 1936, 1949, 1951, 1958, 1975, and 1983 around our study site, which were also documented in the calamity memorandum in Sichuan, China. These may have been related to the extreme high temperatures of these years reconstructed in this study.

### 4.3. Possible Driving Forces for the Temperature Variability in the Southeastern TP

The results of the EEMD analysis indicated the existence of some important cycles in our temperature reconstruction, which may be used to explain the temperature variability in our study region. The EEMD decomposed the original series into seven main intrinsic mode functions (IMF1–IMF7) and one trend (RES) which showed the characteristics of periodic variations on different timescales (Figure 8). Among these intrinsic mode functions, IMF1–IMF5 are highly correlated with the original data ($p < 0.01$), and their rates of contribution to variance are 35.11%, 19.50%, 23.98%, 19.28% and 1.58% (Table 2), respectively. The high correlation coefficients of IMF1–IMF5 with the original dataset suggest that IMF1–IMF5 may better represent the cycle characteristics of the original reconstruction compared to other components.

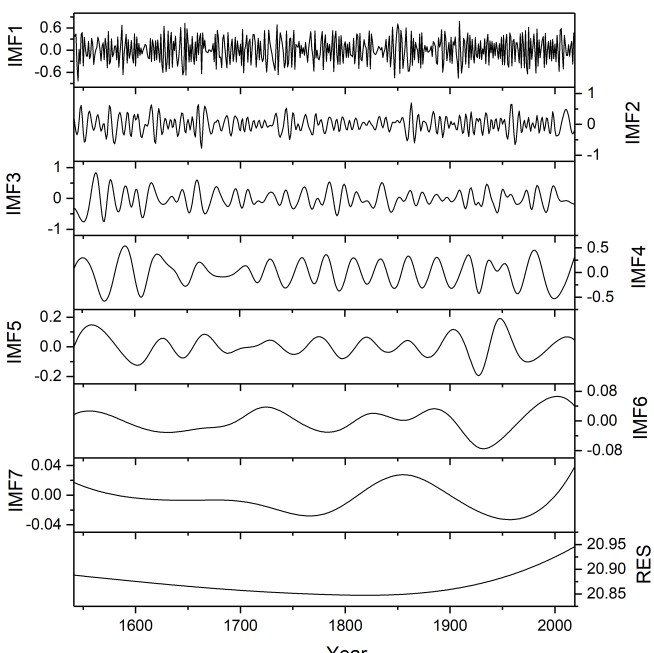

**Figure 8.** Extracted components of reconstructed mean maximum temperature of May–June using EEMD.

The 2–8-year signal shown in IMF1 and IMF2 falls within the spectral bandwidth (2–8 years) of El Nino Southern Oscillation (ENSO) [49,50]. IMF1 and IMF2 account for 54.61% (35.11% + 19.50%) of the total variance. The contribution rate of more than half of the total variance indicates that ENSO may play an important role in temperature changes of the southeastern TP, which has been noted in other previous studies [8,37,38,51]. It has been proven that ENSO can effectively modulate the temperature variability in the tropics and across the globe [52,53]. During the warm phase of ENSO, descending motion over the Indian Peninsula and equatorial ascending motion form the anomalous regional Hadley circulation, surface conditions are drier in this Indian monsoon-controlled study region and temperatures rise above normal [54,55]. Conversely, the cool phase of the ENSO would result in below-normal temperatures.

**Table 2.** Contribution to the variance and coefficient of a reconstruction series after ensemble empirical mode decomposition into six intrinsic mode functions (IMFs) and a trend (RES).

| Variable | IMF1 | IMF2 | IMF3 | IMF4 | IMF5 | IMF6 | IMF7 | RES |
|---|---|---|---|---|---|---|---|---|
| Major cycle (a) | 2.9–4.2 | 4.5–8.3 | 11.1–15.4 | 20–33.3 | 50.4 | 159.7 | 250 | |
| Contribution (%) | 35.11 | 19.50 | 23.98 | 19.28 | 1.58 | 0.30 | 0.09 | 0.15 |
| Correlation coefficient | 0.569 | 0.554 | 0.565 | 0.471 | 0.145 | 0.023 | 0.030 | 0.001 |

The decadal signal of IMF3 is approximately equivalent to the 11-year Schwabe cycles of solar activity which can influence the temperature variations across the globe [56]. Many regional studies have found that 10–12-year cycles of solar activity have an effect on southeastern TP [8,23,57–59].

The cycles of 20–30 years in IMF4 and the cycles of 50.4 years of IMF5 may correspond to the Pacific Decadal Oscillation (PDO), which has often been described as a long-lived El Niño-like pattern of Pacific climate variability [60]. Tree-ring-based temperature reconstructions in surrounding areas, such as the Gaoligong Mountains on the southeastern TP [38], the Animaqin Mountains on the eastern TP [37], the Qilian Mountains on the northeastern TP [61] and the source region of Yangtze River on the eastern TP [25] discussed a possible relationship between the PDO and temperature variability.

The IMF6 and IMF7 components indicate temperature variations on multi-centennial scales, which may be related to the Gleissberg cycle [56]. The RES component shows that there has been a decreasing trend in temperatures since 1541 and an upward trend after 1850. To verify this conclusion, we further conducted a correlation analysis between the reconstructed series with a global SST. As shown in Figure 9, during the period 1962–2019, the significant positive correlation regions were concentrated in the Indian Ocean, equatorial east Pacific and the western north Atlantic ($r > 0.254$, $p < 0.05$). Thus, as can be seen, the spatial correlation mode shows that the temperature variability in southeastern TP may have some relationship with ENSO, PDO and other factors, but the specific mechanism needs to be further analyzed.

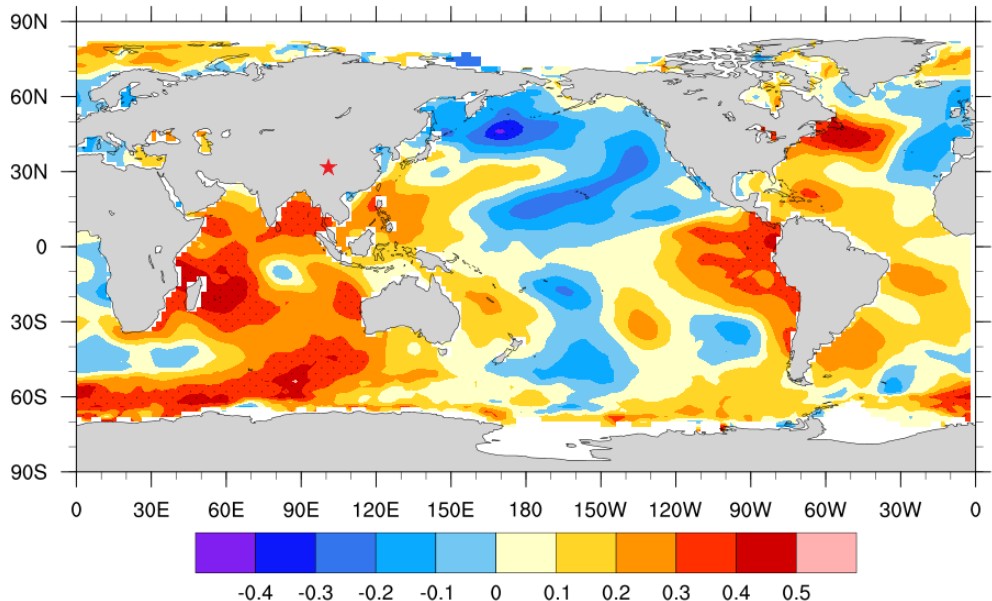

**Figure 9.** The spatial correlation between the reconstructed temperature sequence and global SST during the period 1962–2019. The dotted area passed the confidence test ($p < 0.05$). The pentagram indicates the sampling site.

## 5. Conclusions

In this paper, we presented a 479-year May–June maximum temperature reconstruction for the southeastern TP based on a new tree-ring width chronology of *Picea balfouriana*. The reconstruction of the May–June maximum temperature explained 33.6% of the climatic variance during the calibration period 1962–2019. The reconstruction indicated that no obvious warming trend was observed in our reconstruction for the past 50 years compared to the values for the preceding periods. The spatial correlation analysis and the comparison with other local temperature reconstructions confirmed the reliability and representativeness of our reconstruction. The results of the EEMD analysis showed that the temperature variations in this area may be affected by ENSO cycles, solar activity, and PDO. The correlation analysis between our reconstruction and global sea surface temperatures further proved that the temperature variability in southeastern TP may have some relationship with ENSO, PDO and other factors, but the specific mechanism needs to be further analyzed.

**Author Contributions:** Methodology, J.L.; validation, Y.Z.; formal analysis, Y.Z.; resources, J.L., Z.Z. and S.Z.; data curation, Y.Z.; writing—original draft preparation, Y.Z.; writing—review and editing, Y.Z., Z.Z. and J.L. All authors have read and agreed to the published version of the manuscript.

**Funding:** This research has been funded by National Natural Science Foundation of China Projects (41772173), and the Second Tibetan Plateau Scientific Expedition and Research (STEP) program (grant Nos. 2019QZKK0103 and 2019QZKK0608).

**Data Availability Statement:** Dataset available on request to corresponding author.

**Acknowledgments:** We are truly grateful to the editor and reviewers' comments and thoughtful suggestions.

**Conflicts of Interest:** The authors declare no conflict of interest.

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
