# Peer review of "A 479-Year Early Summer Temperature Reconstruction Based on Tree-Ring in the Southeastern Tibetan Plateau, China"

_atmosphere, doi:10.3390/atmos12101251_

Round 1

Reviewer 1 Report

Interesting tree-ring research in China.  Potentially publishable after revision.

Questions:

Line 98: Detrending method says 50 years of series length smoothing spline.  This is not clear: is it the 50-year smoothing spline, which is pretty flexible, or the smoothing spline that is 50% of the series length, which would be more rigid?

Fig. 3a: I'd add a 1.0 reference line across this plot, to better indicate average growth.

Fig. 3d: The minimum unit of experimentation in dendrochronology is usually the tree, not the core.  Perhaps most trees were cored twice, but Methods says “at least two cores per tree,” so the number of cores per tree might vary.  I’d express sample depth as number of trees.

Back to the question of detrending, the tree-ring chronology (Fig. 3a) looks pretty flat, i.e., little to no low-frequency variation.  If the detrending really was with the 50-year spline, might that have been overly flexible?

For climate modeling, it would be more inclusive to test (show) more multi-month seasons, including of precipitation.  Have you tried Meko’s SEASCORR software?  Meko, D. M., R. Touchan, and K. A. Anchukaitis (2011), Seascorr: a MATLAB program for identifying the seasonal climate signal in an annual tree-ring time series, Computers & Geosciences, 37, 1234-1241.

As anticipated, the reconstruction shows no extremely warm or cold periods during the recent 200 years (lines 169-170), nor any apparent ascending tendency for the past 50 years (lines 173-174).  Could this lack of low-frequency variability be an artifact of overly flexible detrending?

Line 170 says 21-year low-pass filter, but the caption for Fig. 5 says 11-year low-pass filter?

Lines 220-223: There was a warming trend in this reconstruction after the 20th century, but then: The reason why this temperature reconstruction did not show an unprecedented warming in 2000s … .  Which is it?  Yes or no on recent warming in this reconstruction?

Fig. 7: The other reconstructions that extend out to the present (b & d) show much more warming in recent years than this reconstruction does.  Again, it’s concerning that the detrending technique used here might have removed too much variability, namely, decadal-scale and longer variability.

Author Response

Dear reviewer:

We are truly grateful to yours comments and thoughtful suggestions. Based on these comments and suggestions, we have made careful modifications. We hope the new manuscript will meet the journal’s standard. Below you will find our point-by-point responses to the reviewer’s comments/ questions:

Yours sincerely

The comment from reviewer:

Reviewer #1

Interesting tree-ring research in China.  Potentially publishable after revision.

Questions:

1.Line 98: Detrending method says 50 years of series length smoothing spline.  This is not clear: is it the 50-year smoothing spline, which is pretty flexible, or the smoothing spline that is 50% of the series length, which would be more rigid?

Answers: Thanks very much for your suggestion, we are so sorry that we did not explain it clearly in the document because of our negligence. The detrending method we used in this study is the 50-years smothing spline.

  1. Fig. 3a: I'd add a 1.0 reference line across this plot, to better indicate average growth.

Answers: Thanks very much for your suggestion, we have added a 1.0 reference line across Figure 3a.

  1. Fig. 3d: The minimum unit of experimentation in dendrochronology is usually the tree, not the core. Perhaps most trees were cored twice, but Methods says “at least two cores per tree,” so the number of cores per tree might vary. I’d express sample depth as number of trees.

Answers: Thanks very much for your suggestion, we have revised “No. of cores” as “Sample depth”. And the revised figure is as follows:

Figure 3. (a) Tree-ring width index; (b) inter-series correlation (Rbar) for the standard chronology; (c) expressed population signal (EPS) (dashed red line indicates 0.85 threshold); and (d) the sample size (gray shadow) change over time. Critical EPS level of 0.85 is highlighted with vertical blue line.

  1. Back to the question of detrending, the tree-ring chronology (Fig. 3a) looks pretty flat, i.e., little to no low-frequency variation. If the detrending really was with the 50-year spline, might that have been overly flexible?

Answers: Thanks very much for your suggestion, to explore whether the tree-ring chronology detrended by 50-years smoothing spline (CSS50) can objectively reflect the response characteristics of tree growth to climatic conditions in this study area, we compared the chronology detrended by CSS50 with chronologies detrended by other two commonly used detrending methods (cubic smoothing splines at 67% of the series length and negative exponential curve). CSS67% and Neg-ex represent the chronologies detrended by cubic smoothing splines at 67% of the series length and negative exponential curve, respectively.

  • From the low-frequency fluctuation characteristics of each chronology (Figure 1), there is little difference in the low-frequency fluctuation amplitude of CSS67%, CSS50 and Neg-ex chronologies during the reliable period 1541-2019. In addition, each chronology contains many common fluctuation signals, reflecting the interannual variation characteristics of tree growth in the study area. From the correlation characteristics of each chronology, during the reliable period 1541-2019, the correlation between CSS50 chronology and CSS67% chronology was the most obvious (r = 0.937, p < 0.01), followed by CSS50 chronology and Neg-ex chronology (r = 0.903, p < 0.01). The high correlation of these chronologies shows that they have good consistency.

Figure 1. Tree-ring chronologies developed from different detrending methods in southeastern Tibetan Plateau, China. The time span is A.D. 1245 to 2019, inclusive. The red bold smoothed curves superimposed on the annual tree-ring indices emphasize the series of 11-year smoothing average. CSS67%, CSS50 and Neg-ex represent the chronologies detrended by cubic smoothing splines at 67% of the series length, fixed 50 years and negative exponential curve, respectively.

  • According to the correlation analysis results between each chronology and the meteorological data collected from this study area (Figure 2), the response characteristics of CSS50 chronology are very consistent with CSS67% and Neg-ex chronologies except September. Specifically, there is a positive correlation between chronlogies with the temperature (mean, maximum and minimum temperature) in summer (May to July), the most significant correlation with temperature was found in June (p < 0.01), and there also an obvious negative correlation with the temperature in current April (mean and maximum temperature); these chronologies also show good consistency with the correlation characteristics of precipitation, which is mainly reflected in the obvious positive correlation with the precipitation in July of current year, and the negative correlation with may of current year.

Figure 2. Correlation analysis between chronologies derived from different detrending methods and monthly climate data collected from this study area over the period 1962–2019. P and C mark the previous and current year, respectively. The horizontal dashed lines indicate statistically significant correlations at the p < 0.01 level. CSS67%, CSS50 and Neg-ex represent the chronologies detrended by cubic smoothing splines at 67% of the series length, fixed 50 years and negative exponential curve, respectively.

     In summary, The interannual fluctuation characteristics of CSS50 detrend chronology are very similar to those of CSS67% and Neg-ex detrend chronologies, and the correlation between them has reached a high level; in addition, the response characteristics of these chronologies to meteorological variables are very similar. The above discussion shows that the CSS50, CSS67% and Neg-ex detrend chronologies can objectively reflect the response characteristics of tree growth to climate conditions. Our tree ring sample may mainly reflects the characteristics of high frequency climate change in the study region.Therefore, the 50-years smoothing spline we selected in this study did not remove too many low frequency signals.

  1. For climate modeling, it would be more inclusive to test (show) more multi-month seasons, including of precipitation. Have you tried Meko’s SEASCORR software? Meko, D. M., R. Touchan, and K. A. Anchukaitis (2011), Seascorr: a MATLAB program for identifying the seasonal climate signal in an annual tree-ring time series, Computers & Geosciences, 37, 1234-1241.

Answers: Thanks very much for your suggestion, we have tried to do the correlation analysis between multi-month seasons with tree-ring chronlogy before, and the results are shown below. It was found that the highiest correlation was found between the TRW chronology and mean maximum temperatures of May-June. Because showing all the results in the correlation figure is too miscellaneous, we only show the multi-month seasons relevant results in May-June in the document.

Table 1. Correlation analysis between 50-years smothing spline chronology and multi-month seasons collected from this study area over the period 1962–2019.

  1. As anticipated, the reconstruction shows no extremely warm or cold periods during the recent 200 years (lines 169-170), nor any apparent ascending tendency for the past 50 years (lines 173-174). Could this lack of low-frequency variability be an artifact of overly flexible detrending?

Answers: Thanks very much for your suggestion, as we answered in question 4, by comparing with other detrending methods, the 50-years smoothing spline we selected in this study did not remove too many low frequency signals. This lack of low-frequency variability may be attributed to the tree-ring sample in this study area which mainly reflects the characteristics of high frequency climate change.

  1. Line 170 says 21-year low-pass filter, but the caption for Fig. 5 says 11-year low-pass filter?

Answers: Thanks very much for your suggestion, we have revised “21-year low-pass filter” as “11-year low-pass filter” in Line 170.

  1. Lines 220-223: There was a warming trend in this reconstruction after the 20th century, but then: The reason why this temperature reconstruction did not show an unprecedented warming in 2000s … . Which is it? Yes or no on recent warming in this reconstruction?

Answers: Thanks very much for your suggestion, there was indeed a warming trend in this reconstruction after the 20th century, but it was not unprecedented compared with the previous hundreds of years. And the related sentences have been revised to better illustrate the warming trend.

We further calculated the climate tendency rate for each grid point over the region (20°–45°N, 75°–110°E) over the period 1960-2019. According to the figure following, we found that there are a warming trend in our study area but is not significant from the perspective of observation.

Figure 3. Climate tendency rate (monthly maximum temperature data from CRU grids) of TP from 1960 to 2019. The black square denotes our study area.

  1. Fig. 7: The other reconstructions that extend out to the present (b & d) show much more warming in recent years than this reconstruction does. Again, it’s concerning that the detrending technique used here might have removed too much variability, namely, decadal-scale and longer variability.

Answers: Thanks very much for your suggestion, we have explored the cause of different trend showed in this study and other reconstructions. Firstly, the unobvious warming trend in May–June maximum temperature over the past 479 years in our study might be related to the topography and vegetation cover of the region under study. Xinlong is located in an area near to Yalong River, and the elevations of our sampling sites are high and close to treelines. It has been demonstrated that treelines on the southeastern Tibetan Plateau hold much moisture during the rainy season (Liang et al., 2011). This abundant moisture is favorable to mitigate the potential warming by evapotranspiration (Körner, 2003; Holtmeier and Broll, 2005). Secondly, the unobvious warming trend could be attributed to different targets of reconstruction (month/seasons), use of different types of tree-ring proxies (width/density), inconsistent variability in Tmax/Tmean/Tmin (Wilson and Luckman, 2002, 2003). The unobvious warming trend in summer was also reported in other studies from southeastern Tibeatan Plateau (Bräuning and Mantwill, 2004; Linderholm and Brauning, 2006).

 As we answered in question 8, we further calculated the climate tendency rate for each grid point over the region (20°–45°N, 75°–110°E) over the period 1960-2019. According to the figure following, we found that the warming trend in our study area is not significant from the perspective of observation.

Therefore, the unobvious warming trend in this study might irrelevant to the detrending method we used.

Figure 3. Climate tendency rate (monthly maximum temperature data from CRU grids) of TP from 1960 to 2019. The black square denotes our study area.

Reference:

Bräuning, A., Mantwill, B., 2004. Summer temperature and summer monsoon history on the Tibetan Plateau during the last 400 years recorded by tree rings. Geophys. Res. Lett. 31 (24), L24205.

Holtmeier, F.-K., Broll, G., 2005. Sensitivity and response of northern hemisphere altitudinal and polar treelines to environmental change at landscape and local scales. Glob.Ecol. Biogeogr. 14 (5), 395–410.

Körner, C., 2003. Alpine Plant Life: Functional Plant Ecology of High Mountain Ecosystems.

Springer-Verlag, Berlin, Heidelberg.

Liang, E., Liu, B., Zhu, L., Yin, Z.-Y., 2011. A short note on linkage of climatic records between a river valley and the upper timberline in the Sygera Mountains, southeastern Tibetan Plateau. Glob. Planet. Chang. 77 (1–2), 97–102.

Linderholm, H.W., Brauning, A., 2006. Comparison of high-resolution climate proxies from the Tibetan Plateau and Scandinavia during the last millennium. Quat. Int.154, 141–148.

Wilson, R.J.S., Luckman, B.H., 2002. Tree-ring reconstruction of maximum and minimum temperatures and the diurnal temperature range in British Columbia. Canada. Dendrochronologia 20 (3), 257e268.

Wilson, R.J.S., Luckman, B.H., 2003. Dendroclimatic reconstruction of maximum summer temperatures from upper treeline sites in Interior British Columbia, Canada. Holocene 13, 851e861.

Reviewer 2 Report

This study presents a new paleoclimatic reconstruction of summer temperature for the southeastern Tibetan plateau. A tree-ring width chronology was made from 53 cores from two tree species from one site. The chronology correlates well with mean max May-June temperature (r = 0.6). The reconstruction is sufficiently stable and good as some of the statistics usually used in the calibration/verification process in dendrochronology are sufficiently good. The reconstruction is however not able to portrait any mid-term variability of temperatures (for example the past 50 years). The reconstruction show mainly consistency to other temperature reconstitutions from the same region as well as to glacier fluctuations. Links are made with ENSO, solar activity and SST.

If I am usually happy to see published a new tree-ring chronology and climate reconstruction published, I was doubting about the novelties added by this study. I was not convinced by the added value of this reconstruction of temperature comparing to others Keyimu, et al., 2021, Yu, et al., 2018, Duan and Zhang 2014, Huang et al., 2019, Liang et al., 2008) or Wang et al. 2015, Journal of Climate. It is not much longer than other studies for example. The author plaid for an unprecedented maximum temperature reconstruction, but I am not convinced of the novelties of a maximum temperature reconstruction, and I would like the author to better explain the novelty of this reconstruction. If the reconstruction of maximum temperatures is crucial, then could you support it through relationships between the reconstructed warmest years and historical events seen in the archives? That would help to give more strength and interest to this manuscript.

I was also not convinced by the structure of the manuscript; a lot of results are appearing in discussion. The discussion, in turn, suffers from a lack of depth that is quite frustrating to the reader.

Please (1) be more transparent on there objectives and methods choose to answer to objectives, (2) show more impactful results, (3) rewrite the paper to be clearer and discuss more in depth the results.

I have other major concerns that came throughout the text:

L1: You name global warming. But you have not done an analysis to show any warming trend from tree rings, due for example to a not adapted choice of the standardization procedure (see other comments below).

L2: I would not call a regional chronology, a chronology that has been built from trees from a single site (L3).

L54: I don’t believe that you aimed at reconstructing the maximal temperature, but I understand that you choose the climate parameter that correlates the best with the chronology. This is my main concern about the aim and novelty of this study. If you clarify the objectives and you use well-argued methodology (choice of standardization methods, aim of comparing SST and other global cycles with reconstruction, it would greatly help)

L83: Please precise the number of trees (less than 25 trees?).

L82-83: More details about the site would be welcome, as well as the number of spruce and fir trees sampled. Why did you use both species? This is important to know…

L98: I am not convinced by the procedure you use to choose the standardization method. Moreover, I would not have used a 50-years smoothing spline that removes a lot of medium frequency signals. What cut-off value did you use? 0.5? RCS would have been better, but you need more data and dead trees. Use neg-ex and/or very 2/3 of total series length spline. Please choose the method according to your objectives.

Fig. 3: Rbar is quite low after 1500. Any idea why? Was it calculated within cores or within individuals? If it was calculated within cores, then EPS is biased.

L118-120: Why not using CRU dataset instead of meteorological stations to length the period of calibration? It was used for spatial correlations. It would be nice at least to try and see if correlations are high between stations and CRU, and RWI and CRU.

L114: Why leave-one-out cross-verification was used? Why not a cross calibration? Are you able to give a CE? Is the period of calibration too short?

L123: What is the need of EEMD within the aims of your study? What is the added value? Could you discuss more?

L136-142: Ok.

L164-165: Could you relate any of these years to historical events?

L173-175: Maybe due to the standardization? Same for L222

L177-194: Ok for the physiological explanation that makes sense.

L226: Figures for glaciers would be nice

L279 4.4 is basically results. Moreover, I don’t seize the point of showing these results only to conclude on the complexity on the relationships between SST, air temperature and tree ring width. Maybe reconsider showing this, or at least explain why it is interesting. 

Author Response

Dear reviewer:

We are truly grateful to yours comments and thoughtful suggestions. Based on these comments and suggestions, we have made careful modifications. We hope the new manuscript will meet the journal’s standard. Below you will find our point-by-point responses to the reviewer’s comments/ questions:

Yours sincerely

The comment from reviewer:

Reviewer #2

This study presents a new paleoclimatic reconstruction of summer temperature for the southeastern Tibetan plateau. A tree-ring width chronology was made from 53 cores from two tree species from one site. The chronology correlates well with mean max May-June temperature (r = 0.6). The reconstruction is sufficiently stable and good as some of the statistics usually used in the calibration/verification process in dendrochronology are sufficiently good. The reconstruction is however not able to portrait any mid-term variability of temperatures (for example the past 50 years). The reconstruction show mainly consistency to other temperature reconstitutions from the same region as well as to glacier fluctuations. Links are made with ENSO, solar activity and SST.

If I am usually happy to see published a new tree-ring chronology and climate reconstruction published, I was doubting about the novelties added by this study. I was not convinced by the added value of this reconstruction of temperature comparing to others Keyimu, et al., 2021, Yu, et al., 2018, Duan and Zhang 2014, Huang et al., 2019, Liang et al., 2008) or Wang et al. 2015, Journal of Climate. It is not much longer than other studies for example. The author plaid for an unprecedented maximum temperature reconstruction, but I am not convinced of the novelties of a maximum temperature reconstruction, and I would like the author to better explain the novelty of this reconstruction. If the reconstruction of maximum temperatures is crucial, then could you support it through relationships between the reconstructed warmest years and historical events seen in the archives? That would help to give more strength and interest to this manuscript.

I was also not convinced by the structure of the manuscript; a lot of results are appearing in discussion. The discussion, in turn, suffers from a lack of depth that is quite frustrating to the reader.

Please (1) be more transparent on there objectives and methods choose to answer to objectives, (2) show more impactful results, (3) rewrite the paper to be clearer and discuss more in depth the results.

Answers: We are truly grateful to yours comments and thoughtful suggestions. As you suggested, we revised the objectives in this study which would be more accurate. In the conclusion section, we showed clearer results and deleted unnecessary conclusions. We further compared the reconstructed temperature series with recorded historical events, finding that there were corresponding historical documents near the sampling site, increasing the reliability of reconstruction. And the other questions we answered below:

1.L1: You name global warming. But you have not done an analysis to show any warming trend from tree rings, due for example to a not adapted choice of the standardization procedure (see other comments below).

Answers: Thanks very much for your suggestion, the reason why we mentioned the global warming in the introduction is that we want to explain the importance of exploring the climate change of TP in the context of global warming.

  1. L2: I would not call a regional chronology, a chronology that has been built from trees from a single site (L3).

Answers: Thanks very much for your suggestion, we have deleted “regional” in this sentence.

  1. L54: I don’t believe that you aimed at reconstructing the maximal temperature, but I understand that you choose the climate parameter that correlates the best with the chronology. This is my main concern about the aim and novelty of this study. If you clarify the objectives and you use well-argued methodology (choice of standardization methods, aim of comparing SST and other global cycles with reconstruction, it would greatly help)

Answers: Thanks very much for your suggestion, we have revised this paragraph which describes the objectives of this study. And the revised paragraph is as follows: “Here, we presented a chronology, originating from Xinlong county, on the southeastern TP. As shown below, the high sensitivity of the tree-ring width (TRW) chronology to early summer maximum temperature allows us to develop a reliable reconstruction to perceive regional early summer maximum temperature variations during the past five centuries and to investigate possible driving factors that influence the temperature variability on the southeastern TP.”

  1. L83: Please precise the number of trees (less than 25 trees?).

Answers: Thanks very much for your suggestion, we feel so sorry for the wrong description of sampling method in the document. For some trees, we only sampled one core from each tree. And the revised related sentence is as follows: “Increment cores were obtained from each selected healthy tree by an increment borer at breast height.” Furthermore, we have added the description of number of cores we selected in the document: “Eventually, 46 cores from 31 trees were used to develop a ring-width chronology.”

  1. L82-83: More details about the site would be welcome, as well as the number of spruce and fir trees sampled. Why did you use both species? This is important to know…

Answers: Thanks very much for your suggestion, we are so sorry that we did not explain it clearly in the document because of our negligence. Only the picea balfouriana was sampled in the Spruce-Fir mixed stands. The Abies fabri was not sampled. And the revised paragraph is “Tree-ring samples were collected at 3880 m a.s.l. in August 2019 from Spruce-Fir mixed stands in Xinlong (Figure 1). Only picea balfouriana was sampled. The sampling points have slopes ranging from 30 to 40 degrees and are far from areas of anthropogenic activity and disturbance.”

  1. L98: I am not convinced by the procedure you use to choose the standardization method. Moreover, I would not have used a 50-years smoothing spline that removes a lot of medium frequency signals. What cut-off value did you use? 0.5? RCS would have been better, but you need more data and dead trees. Use neg-ex and/or very 2/3 of total series length spline. Please choose the method according to your objectives.

Answers: Thanks very much for your suggestion, we selected the 0.5 cut-off in our study. And to explore whether the tree-ring chronology detrended by 50-years smoothing spline (CSS50) can objectively reflect the response characteristics of tree growth to climatic conditions in this study area, we compared the chronology detrended by CSS50 with chronologies detrended by other two commonly used detrending methods (cubic smoothing splines at 67% of the series length and negative exponential curve) as you suggested. CSS67% and Neg-ex represent the chronologies detrended by cubic smoothing splines at 67% of the series length and negative exponential curve, respectively.

(1)   From the low-frequency fluctuation characteristics of each chronology (Figure 1), there is little difference in the low-frequency fluctuation amplitude of CSS67%, CSS50 and Neg-ex chronologies during the reliable period 1541-2019. In addition, each chronology contains many common fluctuation signals, reflecting the interannual variation characteristics of tree growth in the study area. From the correlation characteristics of each chronology, during the reliable period 1541-2019, the correlation between CSS50 chronology and CSS67% chronology was the most obvious (r = 0.937, p < 0.01), followed by CSS50 chronology and Neg-ex chronology (r = 0.903, p < 0.01). The high correlation of these chronologies shows that they have good consistency.

Figure 1. Tree-ring chronologies developed from different detrending methods in southeastern Tibetan Plateau, China. The time span is A.D. 1245 to 2019, inclusive. The red bold smoothed curves superimposed on the annual tree-ring indices emphasize the series of 11-year smoothing average. CSS67%, CSS50 and Neg-ex represent the chronologies detrended by cubic smoothing splines at 67% of the series length, fixed 50 years and negative exponential curve, respectively.

  • According to the correlation analysis results between each chronology and the meteorological data collected from this study area (Figure 2), the response characteristics of CSS50 chronology are very consistent with CSS67% and Neg-ex chronologies except September. Specifically, there is a positive correlation between chronlogies with the temperature (mean, maximum and minimum temperature) in summer (May to July), the most significant correlation with temperature was found in June (p < 0.01), and there also an obvious negative correlation with the temperature in current April (mean and maximum temperature); these chronologies also show good consistency with the correlation characteristics of precipitation, which is mainly reflected in the obvious positive correlation with the precipitation in July of current year, and the negative correlation with may of current year.

Figure 2. Correlation analysis between chronologies derived from different detrending methods and monthly climate data collected from this study area over the period 1962–2019. P and C mark the previous and current year, respectively. The horizontal dashed lines indicate statistically significant correlations at the p < 0.01 level. CSS67%, CSS50 and Neg-ex represent the chronologies detrended by cubic smoothing splines at 67% of the series length, fixed 50 years and negative exponential curve, respectively.

     In summary, The interannual fluctuation characteristics of CSS50 detrend chronology are very similar to those of CSS67% and Neg-ex detrend chronologies, and the correlation between them has reached a high level; in addition, the response characteristics of these chronologies to meteorological variables are very similar. The above discussion shows that the CSS50, CSS67% and Neg-ex detrend chronologies can objectively reflect the response characteristics of tree growth to climate conditions. Our tree ring sample may mainly reflects the characteristics of high frequency climate change in the study region.Therefore, the 50-years smoothing spline we selected in this study may did not remove too many low frequency signals. And due to the strongest correlation between this 50-years spline chronology with climate variables, we selected in this study eventually.

  1. Fig. 3: Rbar is quite low after 1500. Any idea why? Was it calculated within cores or within individuals? If it was calculated within cores, then EPS is biased.

Answers: Thanks very much for your suggestion, the Rbar is calculated within individuals, and the decrease in Rbar has been noted in previous study in nearby study area (Deng et al., 2017). This decline in Rbar may be attributed to the anomalous atmospheric circulation events (Anhäuser et al., 2020) which could cause a decrease in internal coherency after 1500. Although the Rbar has decreased, it is still significant, and still relatively stable in combination with EPS.

References:

Anhäuser, T., Sehls, B., Thomas, W., Hartl, C., Greule, M., Scholz, D., Esper, J., Keppler, F. (2020) Tree-ring δ2H values from lignin methoxyl groups indicate sensitivity to European-scale temperature changes. Palaeogeography, Palaeoclimatology, Palaeoecology, 546, 109665.

Deng, Y., Gou, X., Gao, L., Yang, T., Yang, M. (2017) Early-summer temperature variations over the past 563 yr inferred from tree rings in the Shaluli Mountains, southeastern Tibet Plateau. Quaternary Research, 81(3), 513-519.

  1. L118-120: Why not using CRU dataset instead of meteorological stations to length the period of calibration? It was used for spatial correlations. It would be nice at least to try and see if correlations are high between stations and CRU, and RWI and CRU.

Answers: Thanks very much for your suggestion, we have tried to use the CRU dataset to conduct the correlation analysis. And most of the meteorological station records over the TP did not

start until the late 1950s, we only use the more reliable period after AD 1960. The results are shown below. Since the correlation between the climate variables with CRU dataset is lower than with meteorological data, we finally selected the meteorological data for correlation analysis.

Table 1. Correlation analysis between the 50-years smothing spline chronology and monthly climate data collected from CRU TS dataset over the period 1960–2019. P and C mark the previous and current year, respectively.

  1. L114: Why leave-one-out cross-verification was used? Why not a cross calibration? Are you able to give a CE? Is the period of calibration too short?

Answers: Thanks very much for your suggestion, we have revised the leave-one-out cross-verification method as split-period calibration/verification analysis in the document and the CE was given in the table 1.

  1. L123: What is the need of EEMD within the aims of your study? What is the added value? Could you discuss more?

Answers: Thanks very much for your suggestion, EEMD was used in this study to analyze the periodicity of the reconstruction series and to explore the law of periodic variations. The EEMD has an advantage in that it can be used to efficiently extract information in both time and frequency domains directly from the integrated variations and trends. And you can directly see the change trend of each cycle.

  1. L164-165: Could you relate any of these years to historical events?

Answers: Thanks very much for your suggestion, we have related some historical events in the section “4.2.4. Historical events”.

  1. L173-175: Maybe due to the standardization? Same for L222

Answers: Thanks very much for your suggestion, we have explored the cause of different trend showed in this study and other reconstructions. Firstly, the unobvious warming trend in May–June maximum temperature over the past 479 years in our study might be related to the topography and vegetation cover of the region under study. Xinlong is located in an area near to Yalong River, and the elevations of our sampling sites are high and close to treelines. It has been demonstrated that treelines on the southeastern Tibetan Plateau hold much moisture during the rainy season (Liang et al., 2011). This abundant moisture is favorable to mitigate the potential warming by evapotranspiration (Körner, 2003; Holtmeier and Broll, 2005). Secondly, the unobvious warming trend could be attributed to different targets of reconstruction (month/seasons), use of different types of tree-ring proxies (width/density), inconsistent variability in Tmax/Tmean/Tmin (Wilson and Luckman, 2002, 2003). The unobvious warming trend in summer was also reported in other studies from southeastern Tibeatan Plateau (Bräuning and Mantwill, 2004; Linderholm and Brauning, 2006).

 We further calculated the climate tendency rate for each grid point over the region (20°–45°N, 75°–110°E) over the period 1960-2019. According to the figure following, we found that the warming trend in our study area is not significant from the perspective of observation.

Therefore, the unobvious warming trend in this study might irrelevant to the detrending method we used.

Figure 3. Climate tendency rate (monthly maximum temperature data from CRU grids) of TP from 1960 to 2019. The black square denotes our study area.

Reference:

Bräuning, A., Mantwill, B., 2004. Summer temperature and summer monsoon history on the Tibetan Plateau during the last 400 years recorded by tree rings. Geophys. Res. Lett. 31 (24), L24205.

Holtmeier, F.-K., Broll, G., 2005. Sensitivity and response of northern hemisphere altitudinal and polar treelines to environmental change at landscape and local scales. Glob.Ecol. Biogeogr. 14 (5), 395–410.

Körner, C., 2003. Alpine Plant Life: Functional Plant Ecology of High Mountain Ecosystems.

Springer-Verlag, Berlin, Heidelberg.

Liang, E., Liu, B., Zhu, L., Yin, Z.-Y., 2011. A short note on linkage of climatic records between a river valley and the upper timberline in the Sygera Mountains, southeastern Tibetan Plateau. Glob. Planet. Chang. 77 (1–2), 97–102.

Linderholm, H.W., Brauning, A., 2006. Comparison of high-resolution climate proxies from the Tibetan Plateau and Scandinavia during the last millennium. Quat. Int.154, 141–148.

Wilson, R.J.S., Luckman, B.H., 2002. Tree-ring reconstruction of maximum and minimum temperatures and the diurnal temperature range in British Columbia. Canada. Dendrochronologia 20 (3), 257e268.

Wilson, R.J.S., Luckman, B.H., 2003. Dendroclimatic reconstruction of maximum summer temperatures from upper treeline sites in Interior British Columbia, Canada. Holocene 13, 851e861.

  1. L226: Figures for glaciers would be nice

Answers: Thanks very much for your suggestion. Due to the literatures we used to validate the reconstruction are about different glaciers, here exist some non-coincident advance and retreat periods. Thus, we use the texts instead of figures to describe the glacier advance and retreat periods which are consistent with our reconstruction.

  1. L279 4.4 is basically results. Moreover, I don’t seize the point of showing these results only to conclude on the complexity on the relationships between SST, air temperature and tree ring width. Maybe reconsider showing this, or at least explain why it is interesting.

Answers: Thanks very much for your suggestion, we have deleted the section 4.4 and added the correlation analysis between SST and reconstructed temperature to the section 4.3. The purpose of this correlation analysis is to prove that the temperature variability in southeastern TP may have some relationship with ENSO, PDO and other factors.

Reviewer 3 Report

Comments and Suggestions for Authors

This is the review of the manuscript  (Manuscript ID: atmosphere-1336264)

Journal: MDPI Atmosphere

Submitted to section: Climatology,

Authors: Yu Zhang, Jinjian Li, Zeyu Zheng, Shenglan Zeng

Title: A 479-year early summer temperature reconstruction based on tree-ring in the southeastern Tibetan Plateau, China

Authors reconstruct May-June maximum temperature over the period 1541-2019 based on tree-ring of Spruce-Fir forest.

I have few comments and suggestions to authors.

Below I list specific comments:

Interesting article, although there are already many reconstructions of meteorological and hydrographic elements from Tibet.

abstract: line 7, don't call the May-June period a year but a season or a period

introduction line 53-54 Objective 2 to find the main climatic factor and objective 3 is to indicate it - it cannot be like this, correct objective 3 - do not indicate exactly what you are reconstructing

line 50 an unnecessary hyphen in varia-tions

figure 1 the research area is shown as a rectangle on the China map and in the main figure it is shown as a square ... ??

scale marker - the unit should be selected in such a way that there are no incomplete values, such as 62.5 km here (but e.g. 100 km and 50 km), signature: unnecessary capital letter for Pentagram

figure 2 - can the whole word in the legend be called Precipitation?

2.2. Climate data (Climate in capital letters)

line 72 incorrect indication of longitude 100 E0

line 82: give the exact Latin and English names of the species tested, how many trees of each species were tested, how the samples were taken in relation to the slope

line 90-91 - how many samples were there with missing rings

line 100 - how many samples make up the whole chronology (how many from what genre), what is the period of this chronology, the average annual increase, etc.

figure 3 the marking of the blue vertical line should be transferred to figure 3c

Fig. 4. explain in the caption what P10, P ..., C1, C .. and C56 mean

lines 166 and 167 do not named two or three months as a year

line 164 and figure 5b - it should be the same in the text and in the figure - it is +/- 1.5 SD and the figure is +/- 1 SD

figure 6 mark the research location with a star on both figures (as in figure 1)

Table 2. Major cycle (a), what does (a)  mean?, can say that it is about years

figure 9 mark the research location with a star on all figures (as in figure 1)

Data availability statement: it is a good idea to include the chronology in an open data repository, under an open license with attribution, e.g. CC BY 4.0

Author Response

Dear reviewer:

We are truly grateful to yours comments and thoughtful suggestions. Based on these comments and suggestions, we have made careful modifications. We hope the new manuscript will meet the journal’s standard. Below you will find our point-by-point responses to the reviewer’s comments/ questions:

Yours sincerely

The comment from reviewer:

Reviewer #3

Authors reconstruct May-June maximum temperature over the period 1541-2019 based on tree-ring of Spruce-Fir forest. I have few comments and suggestions to authors.

Below I list specific comments:

Interesting article, although there are already many reconstructions of meteorological and hydrographic elements from Tibet.

1.abstract: line 7, don’t call the May-June period a year but a season or a period.

Answers: Thanks very much for your suggestion, we have revised the sentence “There were 34 extremely warm years (7.2% of total years) and 36 extremely cold years (7.5% of 8 total years) during the reconstruction period.” as “It showed that there were 34 extremely warm summers (7.2% of total summers) and 36 extremely cold summers (7.5% of total summers) during the reconstruction period.”

  1. introduction line 53-54 Objective 2 to find the main climatic factor and objective 3 is to indicate it - it cannot be like this, correct objective 3 - do not indicate exactly what you are reconstructing.

Answers: Thanks very much for your suggestion, we have revised this part as “As shown below, the high sensitivity of the tree-ring width (TRW) chronology to early summer maximum temperature allows us to develop a reliable reconstruction to perceive regional early summer maximum temperature variations during the past five centuries and to investigate possible driving factors that influence the temperature variability on the southeastern TP.”

  1. line 50 an unnecessary hyphen in varia-tions.

Answers: Thanks very much for your suggestion, we have deleted the unnecessary hyphen.

  1. figure 1 the research area is shown as a rectangle on the China map and in the main figure it is shown as a square ... ??scale marker - the unit should be selected in such a way that there are no incomplete values, such as 62.5 km here (but e.g. 100 km and 50 km), signature: unnecessary capital letter for Pentagram.

Answers: Thanks very much for your suggestion, we have revised Figure 1 as you suggested. The study area is shown as a square on the China map and the scale marker has been revised. The unnecessary capital letter for Pentagram in the figure caption has been revised in lower case. And the revised figure is as follows:

Figure 1. Tree-ring sampling site (pentagram) and three meteorological stations (triangle) in the southeastern Tibetan Plateau.

  1. figure 2 - can the whole word in the legend be called Precipitation?

Answers: Thanks very much for your suggestion. The words are too long and if the whole words are used, they will coincide with the lines and destroy the beauty of the picture. Therefore, abbreviations are used and the meaning of “Pre” has been explained in the figure caption.

  1. 2.2. Climate data (Climate in capital letters).

Answers: Thanks very much for your suggestion, we have revised the “climate data” as “Climate data”.

  1. line 72 incorrect indication of longitude 100 E0.

Answers: Thanks very much for your suggestion, we have revised the “100 E°” as “100 ° E” in the document.

  1. line 82: give the exact Latin and English names of the species tested, how many trees of each species were tested, how the samples were taken in relation to the slope.

Answers: Thanks very much for your suggestion, we are so sorry that we did not explain it clearly in the document because of our negligence. Only the picea balfouriana was sampled in the Spruce-Fir mixed stands. The Abies fabri was not sampled. And the revised paragraph is “Tree-ring samples were collected at 3880 m a.s.l. in August 2019 from Spruce-Fir mixed stands in Xinlong (Figure 1). Only picea balfouriana was sampled. The sampling points have slopes ranging from 30 to 40 degrees and are far from areas of anthropogenic activity and disturbance.”

  1. line 90-91 – how many samples were there with missing rings.

Answers: Thanks very much for your suggestion, we have added the description of the number of samples with missing rings in the document “Sample cores with abnormal ring features, such as missing rings, which are difficult to cross-dated, were exclude (7 samples with missing rings).”

  1. line 100 - how many samples make up the whole chronology (how many from what genre), what is the period of this chronology, the average annual increase, etc.

Answers: Thanks very much for your suggestion, we have added these contents as you suggested in section “2.3. Tree-ring data and chronology development”.

  1. figure 3 the marking of the blue vertical line should be transferred to figure 3c.

Answers: Thanks very much for your suggestion, the blue vertical line in the Figure 3 denotes the beginning of reliable period in all subgraphs of Figure 3, so we are so sorry that it can not be transferred to figure 3c.

  1. Fig. 4. Explain in the caption what P10, P …, C1, C .. and C56 mean.

Answers: Thanks very much for your suggestion, we have explained the meaning of P10, P …, C1, C .. and C56 in the figure caption as “P marks the previous year and C marks the current year. The label C56 denotes the average during May–July.”

  1. lines 166 and 167 do not named two or three months as a year.

Answers: Thanks very much for your suggestion, we have revised these sentences as “According to our reconstruction, there were 34 extremely warm summers (7.2% of total summers) and 36 extremely cold summers (7.5% of total summers) during the reconstruction period. The warmest summer (1659) was 22.67 °C and the coldest summer (1606) was 19.09 °C.”

  1. line 164 and figure 5b – it should be the same in the text and in the figure – it is +/- 1.5 SD and the figure is +/- 1 SD.

Answers: Thanks very much for your suggestion, we have revised Figure 5b as you suggested. And the revised figure is as follows:

Figure 5. Comparison between the reconstructed and observed temperature during the period 1962–2019 (a); the reconstructed mean maximum temperatures in May-June using the reliable pe-riod of the TRW chronology from 1541 to 2019 (b).

  1. figure 6 mark the research location with a star on both figures (as in figure 1).

Answers: Thanks very much for your suggestion, we have marked the research location with a star on all figures in Figure 6. And the revised figure is as follows:

Figure 6. Spatial correlations of (a) observed and (b) reconstructed mean maximum May-June temperatures for the southeastern Tibetan Plateau with gridded June–July temperature from Cli-matic Research Unit (CRU) TS 4.04 [26] for the period 1962–2019. The pentagram indicates the sampling site.

  1. Table 2. Major cycle (a), what does (a) mean?, can say that it is about years

Answers: Thanks very much for your suggestion, the “(a)” means year, and we have revised “(a)” as “(year)” in Table 2.

  1. figure 9 mark the research location with a star on all figures (as in figure 1).

Answers: Thanks very much for your suggestion, we have marked the research location with a star on all figures in Figure 9. And the revised figure is as follows:

Figure 9. The spatial correlation between the reconstructed temperature sequence and global SST over the period: 1962-2019. The dotted area passed the confidence test (p < 0.05). The pentagram indicates the sampling site.

  1. Data availability statement: it is a good idea to include the chronology in an open data repository, under an open license with attribution, e.g. CC BY 4.0.

Answers: Thanks very much for your suggestion, we’ll consider including the chronology in an open data repository after the paper is received.

Round 2

Reviewer 2 Report

I acknowledge an improvement of the paper of Zhang et al., and I think the paper should be published.

However, I recommend a more thorough re-read of grammar and typos. Many times, I saw hyphens that should not appear. Please verify each hyphen that appears in the text. L373 -> correlation instead of correlaton. The english of recently written parts is not always clear, please re-read carefully: L88 (I understand what you mean, but "points" can not have "a slope". The sampled trees are growing on 30-40% slopes, or something like this).

I appreciate the discussion of the low CE value of your reconstruction. Indeed, this low value indicates a loss of reconstruction skill, maybe in the lower frequencies as the tree-ring chronology does not show much variation in the low frequencies, that you explained. 

I appreciate the added part about historical archives. Could you create a figure showing the cold and the dry events directly on the reconstructed temperature series? You could add it in supplementary material.

Author Response

Dear reviewer:

We are truly grateful to yours comments and thoughtful suggestions. Based on these comments and suggestions, we have made careful modifications. We hope the new manuscript will meet the journal’s standard. Below you will find our point-by-point responses to the reviewer’s comments/ questions:

Yours sincerely

The comment from reviewer:

Reviewer #2

I acknowledge an improvement of the paper of Zhang et al., and I think the paper should be published.

However, I recommend a more thorough re-read of grammar and typos. Many times, I saw hyphens that should not appear. Please verify each hyphen that appears in the text. L373 -> correlation instead of correlaton. The english of recently written parts is not always clear, please re-read carefully: L88 (I understand what you mean, but "points" can not have "a slope". The sampled trees are growing on 30-40% slopes, or something like this).

I appreciate the discussion of the low CE value of your reconstruction. Indeed, this low value indicates a loss of reconstruction skill, maybe in the lower frequencies as the tree-ring chronology does not show much variation in the low frequencies, that you explained.

I appreciate the added part about historical archives. Could you create a figure showing the cold and the dry events directly on the reconstructed temperature series? You could add it in supplementary material.

We are truly grateful to yours comments and thoughtful suggestions. And the other questions we answered below:

\

  1. However, I recommend a more thorough re-read of grammar and typos. Many times, I saw hyphens that should not appear. Please verify each hyphen that appears in the text.

Answers: Thanks very much for your suggestion, we have re-read the paper and deleted hyphens that should not appear. And the grammar has been checked.

  1. L373 -> correlation instead of correlaton. The english of recently written parts is not always clear, please re-read carefully: L88 (I understand what you mean, but "points" can not have "a slope". The sampled trees are growing on 30-40% slopes, or something like this).

Answers: Thanks very much for your suggestion, the “correlaton” has been revised as “correlation”. And the related description of slopes of tree-ring sampling site has been revised as you suggested. “The sampled trees are growing on 30-40$\%$ slopes and are far from areas of anthropogenic activity and disturbance.”

  1. I appreciate the added part about historical archives. Could you create a figure showing the cold and the dry events directly on the reconstructed temperature series? You could add it in supplementary material.

Answers: Thanks very much for your suggestion, we have added the cold and the dry events directly on the reconstructed temperature series in Figure 5b. The revised figure is shown below:

Figure 5. Comparison between the reconstructed and observed temperature during the period 1962–2019 (a); the reconstructed mean maximum temperatures in May-June using the reliable period of the TRW chronology from 1541 to 2019 with historical extreme cold (blue circle) and the extreme dry (red circle) events corresponding to it (b).
